# Fruit and Vegetable Consumption Patterns and Risk of Chronic Diseases of Lifestyle among University Students in Kenya

**DOI:** 10.3390/ijerph19126965

**Published:** 2022-06-07

**Authors:** Moriasi Abednego Nyanchoka, Martha Elizabeth van Stuijvenberg, Ayuk Betrand Tambe, Mthokozisi Kwazi Zuma, Xikombiso Gertrude Mbhenyane

**Affiliations:** 1Division of Human Nutrition, Faculty of Medicine and Health Sciences, Stellenbosch University, Cape Town 8000, South Africa; abednegomoriasi@yahoo.com (M.A.N.); ayuk.betrand@yahoo.com (A.B.T.);; 2Non-Communicable Diseases Research Unit, South African Medical Research Council, Cape Town 8000, South Africa; lize.vanblerk@gmail.com

**Keywords:** fruits, vegetables, fruit and vegetable consumption, chronic diseases of lifestyle, young adults, Kenya, Africa

## Abstract

This study assesses the fruit and vegetable consumption patterns and risk factors for chronic diseases of lifestyle (CDL) among young adults attending Kenyatta University, Kenya. Four hundred and twenty-three young adults aged 19–30 years participated in the cross-sectional study. Males were 75.0% less likely to be overweight than females (OR = 0.25; 0.13–0.47). The mean overall consumption of fruit and/or vegetables amounted to 3.6 servings in a typical day. Eight in ten participants consumed lower amounts of fruits and vegetables than recommended by the WHO. At least one combined risk factor of CDL was observed among 91.3% of the young adults. About 8.7% of the participants were classified as low risk for CDL (having none of the five risk factors), 48.9% had one risk factor, 32.9% had two risk factors, 8.5% had three risk factors, while 0.9% had four risk factors. A significant relationship between daily fruit consumption and waist circumference was observed. These findings show the dire need to develop public health nutrition activities aimed at increasing the intake of fruit and vegetables and sensitizing young adults, and the general population, to the risk factors of CDL.

## 1. Introduction

In the past few decades, chronic diseases of lifestyle (CDL) such as cardiovascular diseases, cancers, type 2 diabetes, and obesity are affecting younger populations at an alarmingly higher rate than before. The World Health Organization (WHO) projects that by 2030, the global burden of disease attributable to CDL will increase to 56% from 49% in 2005, with the most significant increase expected in the African and East Mediterranean regions. This epidemiologic transition has been attributed to unhealthy lifestyles such as unhealthy diets, physical inactivity, tobacco smoking, and the harmful use of alcohol adapted by this population [1,2,3]. The nutrition transition and changes in lifestyle have contributed to the significant early onset of CDL in the adult population at a younger age in developing countries than in developed countries [2,4,5]. Kenya is increasingly faced with a dietary shift as a result of the consumption of foods that are highly processed, high in calories, sugars, and salt, and low consumption of fruit, vegetables, legumes, and whole grains, and termed to have poor overall diet quality and unhealthy patterns, which includes very low consumption of fruit and vegetables in the general population [5]. Poor intake of fruit and vegetables among young adults have been reported worldwide [6,7,8,9]. 

The connection between the intake of fruit and vegetables and overall population health is well documented, and cannot be underestimated [5,10,11,12,13,14,15,16,17,18]. The World Health Organization (WHO) emphasizes the inclusion of fruits and vegetables in our diets to promote health and lower the risk of certain non-communicable diseases (NCDs), [19,20]. Despite an increasing focus on the importance of the adequate intake of fruit and vegetables, the WHO reports a low prevalence of consumption has been observed worldwide and estimated that 3.9 million global deaths in 2017 were attributable to inadequate consumption of fruit and vegetables [21]. A gap still exists between the recommended and actual intake, and huge populations worldwide are not meeting the quantity and variety that they should consume [5,17,18,22]. 

Fewer studies have focused on the consumption of fruit and vegetables in young adulthood. It is well documented that the transition into young adulthood plays a vital role in the adoption of dietary and lifestyle choices that could be carried to later adulthood, which can have health implications in later life [7,23], and could help prevent or delay the onset of CDL in young adulthood or later in life [24]. This period is characterized by lifestyle changes, i.e., leaving home, leaving familiar context, joining university or college, starting work, and increased independence and autonomy in decision making. As autonomy increases, young people are regularly challenged to make healthy food choices, making them vulnerable to poor dietary habits, high consumption of fast foods and high-calorie foods, which can potentially lead to increased risk to CDL [6,24,25,26]. Despite the weight of evidence on the important role this transition plays in the adoption of a healthy lifestyle (healthy eating patterns and physical activity), there is still scant information on fruit and vegetable consumption patterns in young African adults.

Several studies assessing the consumption of fruit and vegetables worldwide have indicated few young adults are meeting the recommended intake levels [17,18,24,25,27,28,29]. However, little is known about the levels and adequacy of fruit and vegetable consumption patterns, and their association with risk factors of CDL in young adults, in Kenya and the rest of Africa. In view of the above, the aim of this study was to assess the fruit and vegetable consumption patterns and risk factors of CDL in young adults. This will provide a basis to develop suitable intervention programmes that encourage adequate consumption of fruit and vegetables and address the growing burden of CDL in young adulthood. 

## 2. Methods

### 2.1. Study Design

This study used a cross-sectional analytical design to measure the prevalence of fruit and vegetable consumption, risk factors for CDL, and identify risk factors associated with fruit and vegetable intake among young adults attending Kenyatta University (KU) in Kenya. The cross-sectional design was used since the study focused on collecting data at one point in time and used a section of the study population [30]. Moreover, it is relatively simple and inexpensive, and is suitable for conducting a preliminary investigation [30] to look at the relationship between fruit and vegetable consumption and the risk factors of chronic disease of lifestyle, since data in this study area is scant in university students in Kenya. 

### 2.2. Study Population

The study used a regionally representative sample, with all Kenyan regions represented, of young adults aged 19 to 30 years, attending Kenyatta University in Kenya. This was a single-centre study of students at Kenyatta University’s main campus registered for the 2019/2020 academic year. The main campus had 56,405 active students registered for the 2019/2020 academic year according to the Kenyatta University Registrar Academic Office. The main campus has the most diverse profile of all the campuses of the university, including students of both genders, various ethnicities, and its proximity to Nairobi providing an urban lifestyle. Participants were enrolled on the study in June 2019. They were included if they were registered students attending Kenyatta University’s main campus, aged 19 to 30 years, English and/or Swahili speaking, and considered healthy. Participants were excluded if they were under 19 or above 30 years, sick, pregnant, or failed to consent to the study.

### 2.3. Sampling

The size of the sample was determined using Open-Epi version 3.01 (Open Epidemiological statistics for public health) statistical software with the following assumptions: population size of students in Kenyatta University main campus on April 2019 was 56, 405; with a hypothesised 6% frequency of fruit and vegetable consumption derived from the Kenya STEPwise survey; a 95% level of confidence, 5% margin of error and a design effect of 1.0 (all participants were selected from one university), bringing the total study size to 382 participants. The study population was oversampled by 10% bringing the final total study population size to 423 participants. 

Sampling was multistage and included a screening phase and final selection phase. Students were screened prior to study enrolment based on inclusion and exclusion criteria. Those meeting inclusion (being a registered student attending Kenyatta University main campus, aged 19 to 30 years, who self-reported to be healthy and could speak English and/or Swahili languages) were stratified into 19 strata representing each school at Kenyatta University main campus. Participants were excluded from the study if they were pregnant at the time of the study or did not give consent. During the final selection phase, simple random sampling was utilized to select the final study group and to ensure representation of the target population. Microsoft Excel (version 16.41) was used to generate random numbers from the study recruitment list to achieve simple random sampling. 

### 2.4. Data Collection

The data collection phase was conducted for eight weeks in July, August, and September 2019. Participants selected to participate in the main study were contacted for their participation at the data collection centre/location at the nutrition skills laboratory between 9:00 am and 4:00 pm. Participants’ availability was considered when allocating a suitable date and time to attend data collection. 

Before data collection, the principal researcher and research assistants described the study and its aims to the participants. The concepts of voluntary participation and confidentiality were explained to the participants, and the principal researcher and research assistants responded to any questions or concerns raised. A consent form was completed, and a copy in a local language, Swahili, was given to the participants for their records. Each participant spent 30 to 60 min completing all phases of data collection. Each participant was gifted with a bag of fruit after completing data collection as a token of appreciation for their participation.

The study data collection tools and structure were adapted from the validated WHO STEPwise approach to chronic diseases risk factor surveillance tool [31] with modification to suit the Kenyan situation and study. The WHO STEPS tool has been utilized in several surveys to collect data and measure NCD risk factors, and cover three main areas relevant to this study: behavioural risk factors, biological risk factors, and physical measurements. Tools content and context were also assessed in the process of ethical approval by two ethical review committees. 

The questionnaires were self-administered and divided into four sections: sociodemographic characteristics, behavioural risk, food frequency questionnaire, and 24 h diet recall. The first section of the questionnaire covered general sociodemographic information and provided information on age, gender, place of living, and household size and income. The second part provided information on biological risk, history of chronic diseases of lifestyle provided information on participants and family history of CDL. The biological risk was also evaluated using physical and biochemical measurements. The third section of the questionnaire was used to obtain data on behavioural risk; for participants’ current or past health behaviours associated with the risk factors for CDL. These included information on tobacco use, alcohol consumption, fruit and vegetable consumption, physical activity, sedentary behaviour, and knowledge on the importance of fruit and vegetable consumption on an individual’s health and the participants’ practices. To assist with an accurate estimation of fruit and vegetable servings/portion size, each participant was provided with a food album and dietary aids (measuring cups). Where needed, the researcher and assistants were available to assist the participants.

Physical measurements (height, weight, body composition, waist circumference, hip circumference, and blood pressure) and biochemical measurements (random blood sugar and total random cholesterol) were taken and recorded for all participants. Anthropometric and biochemical assessments took place in separate and private rooms (skills laboratory) to ensure privacy and dignity were reserved. The measurements were taken by a trained nutritionist and dietician. A priori standards of procedures protocols were also utilised during the data collection phase. All anthropometric and biochemical equipment was calibrated at the start of each day of data collection.

Height was measured on a flat, stable surface, using a platform and a handheld digital portable ultrasound stadiometer (InBody PUSH stadiometer In Lab 250), which uses an ultrasound sensor to measure an individual’s height with pinpoint accuracy, and recorded in centimetres. A digital, bioelectrical impedance analysis (BIA) scale (Tanita BC-731) was used to measure weight and body composition. The waist and hip circumference were measured using a Seca 150 cm waist circumference tape measure with a tensioning device. 

Waist circumference (WC) measurements for central obesity were classified and interpreted according to the International Diabetes Federation (IDF) criteria with ethnic-specific values for waist circumference [32]. The waist–height ratio (WHtR) was computed by dividing WC (cm) by height (cm) and was categorized as follows: <0.5 (appropriate) and ≥0.5 (increased risk) for metabolic and cardiovascular disease. The waist–hip (WHR) ratio was classified and interpreted according to WHO criteria [33]. Women whose waist-to-hip ratio is ≥0.85 and men with a WHR ≥ 0.9 are centrally obese, and at increased risk of obesity-related morbidity. 

Blood pressure (BP) was measured for all participants using a battery-powered digital blood pressure machine. The definition and classification of hypertension was adopted from the Kenya National Guidelines for Cardiovascular Disease Management. 

The biochemical assessment in this study included random blood glucose and total random cholesterol and were measured using a Nesco MultiCheck IVD machine (NW01H). Random blood sugar was classified and interpreted using the diagnostic criteria provided by the Kenya National Guidelines for the Management of Diabetes Mellitus. Random blood glucose values below 7.8 mmol/L were considered normal, values between 7.8–11.0 mmol/L were considered pre-diabetes stage, and values above 11.1 mmol/L were considered as diabetes. There are no standard cut-off points for total cholesterol levels for the general healthy population. The current study used cut-off values used by the Kenya STEPwise survey [2], where values below 5.0 mmol/L were considered normal, while values above 6.2 mmol/L were considered high. 

### 2.5. Data Analysis

Data were captured on pre-prepared Microsoft Excel version 16.41 (Microsoft Corp, Redmond, WA, USA) spreadsheets at the end of the data collection period. A statistical package for the social sciences (IBM SPSS for Mac version 26, SPSS Inc. Chicago, IL, USA) was used to analyse the quantitative data. All the data were summarised using descriptive statistics. The mean and standard deviation were used to describe data. Binary logistic regression was used to establish associations between fruit and vegetable intake and risk factors of CDL, and the likelihood of having a risk factor of chronic disease of lifestyle in relation to the consumption of fruit and vegetables and the unadjusted odds ratios (OR) and 95% confidence intervals (CI) were calculated. A *p*-value of <0.05 represented statistical significance in hypothesis testing and a confidence interval of 95% was used to describe the estimation of unknown parameters. 

## 3. Results

### 3.1. Demographics

A total of 423 individuals were included in the study, with 39.7% males and 60.3% females (Table 1). Age distribution of the study sample ranged from 19 years to 30 years with a mean age of 22.46 (SD = 2.96) years. Most (79.7%) of the participants were between 19 and 24 years. The sociodemographic characteristics of the study participants are presented in Table 1.

### 3.2. Prevalence of Risk Factors

#### 3.2.1. Biological Risk Factors

More than half (53.2%) of the sample reported having a family history of high blood pressure, while 17.3% reported family history of raised cholesterol, and 31.9% reported having family members with diabetes mellitus. Fourteen percent reported having had a family history of cancer, 11.1% reported having a family member diagnosed with a stroke, and 7.3% reported a family history of heart disease.

The study sample had a mean height of 173.3 cm for males and 162.2 cm for females, mean weight of 62.9 kg for males and 60.3 kg for females, with a mean of body mass index (BMI) of 20.9 kg/m^2^ for males and 22.9 kg/m^2^ for females (Table 2). The overall prevalence of overweight and obesity in the study sample was 13.9% and 4.3%, respectively. Males were 75.0% less likely to be overweight than females (OR = 0.25; 0.13–0.47) with statistical significance (*p* = 0.0001). The overall, mean total body fat percentage for males was 14.1% and 30.4% for females (Table 2). Males were 66% less likely to be overweight than females (OR = 0.34; 0.21–0.56; *p* = 0.0001). 

A statistically significant relationship was observed between gender and waist circumference classification (*p* = 0.0001), and males were 93.0% less likely to have central obesity than females (OR = 0.07, 0.03–0.20). Eighteen percent of the participants had a WHtR ≥ 0.5, and this was comparatively higher among females (25.5%,) than males (7.7%). A statistically significant relationship was found between gender and WHtR (*p* = 0.0001). Further analysis revealed that males were 75.0% less likely to have a higher WHtR than females (OR = 0.25; 0.13–0.46).

#### 3.2.2. Blood Pressure

A higher systolic blood pressure (SBP) was observed among males (123 mmHg) than among females (115 mmHg). A statistically significant difference between males’ and females’ mean systolic blood pressure was observed (*p* = 0.0001). A statistically significant difference was observed between the mean diastolic blood pressure of males and females, with a higher mean for males (77.9 mmHg) than females (75.1 mmHg), with a *p*-value of 0.001 (Table 2).

The results show that 8.3% of the participants had raised blood pressure (SBP ≥ 140 and/or DBP ≥ 90 mmHg). Table 2 shows that a higher proportion of male participants had raised blood pressure (13.7%) than females (4.7%). Males were 3.2 times more likely to have raised blood pressure than females (OR = 3.21; 1.55–6.65) (*p* = 0.002). Approximately nine in ten females had normal blood pressure compared with six in ten males. Two percent (2.6%) of the participants had isolated systolic hypertension. Overall, 1.0% of the participants were diagnosed with severe hypertension (SBP ≥ 160 mmHg and/or DBP ≥ 100 mmHg). 

#### 3.2.3. Biochemical Measurements

The mean random blood glucose for all participants was 4.8 mmol/L. Only 0.5% of the participants had a borderline raised random blood glucose of 7.8–11.1 mmol/L. The mean total cholesterol for all participants was 4.7 mmol/L, with a statistically significant difference (*p* = 0.0001) between the mean of males (4.9) and females (4.6). Overall, more than a third (39.5%) of the participants had raised total cholesterol (≥5.0 mmol/L). More than half (51.2%) of male participants had raised cholesterol, compared with nearly a third (31.8%) of the female participants. Four percent of the participants had high total cholesterol (≥6.2 mmol/L), with a significantly higher prevalence among male participants (8.9%) compared with female participants (0.8%). Further analysis related raised total cholesterol with gender (*p* = 0.0001) and revealed that males were 2.3 times more likely to have raised total cholesterol than females (OR = 2.3; 1.5–3.4).

#### 3.2.4. Behavioural Risk Factors

The current daily tobacco use is relatively low in this sample, with a prevalence of 2.8% and a significantly higher prevalence among males (6.0%) than females (0.8%) (*p* = 0.008). Daily tobacco smoking is 8.01 times higher among males than females (OR = 8.01; 95% CI 1.73–37.02). Analysis indicates that 28.1% of the participants are current alcohol consumers, with more males (33.3%) than females (24.7%). Male participants were 1.52 times as likely as females to be current alcohol consumers (OR = 1.52; 95% CI 0.99–2.34). Excessive alcohol consumption (≥60 g of pure alcohol in men and ≥40 g of pure alcohol in women) was observed in 8.3% of the male participants and 4.7% of the females, who had consumed six or more drinks on a single occasion in the past 12 months. Overall, participants spend an average of 182 min per day in total physical activity with males spending more time (mean minutes 251.6) than females (mean minutes 136.3). A statistically significant difference was observed between gender and the mean minutes of total physical activity (*p* = 0.0001). 

The levels of physical activity were based on a combination of metabolic equivalent (MET) minutes of moderate- and vigorous-intensity activity in a week. Using this combination, overall, 36.2% of the participants had lower physical activity (<600 METs), 41.4% were moderately active (600–3000 METs), while 22.5% had a high level of physical activity (>3000 METs). More female participants (43.1%) had lower physical activity levels compared with males (25.6%, n = 43). Gender was statistically related to the level of physical activity, with a *p*-value of 0.0001.

Overall, participants spend an average of 52 min per day on recreation-related activity. Male participants spend more time (58.7 min) compared with females (42.5 min) per day. Fifty-two percent (52.2%) of the study sample participated in recreation-related physical activities, with a higher percentage (41.6%) participating in moderate work-related physical activity than vigorous work-related physical activities (36.6%). More than half (53.0%) of the male participants participate in vigorous-intensity recreational activities compared with 25.9% female participants. Male participants were 3.23 times more likely than females to participate in vigorous-intensity work-related activities (OR = 3.23; Cl 2.14–4.87), with a statistical significance (*p* = 0.0001). 

Overall participants spend an average of 339 min per day on sedentary time. The median minutes of total sedentary activity in a typical day were 300. A statistically significant association was found between time spent in sedentary activities and gender (*p* = 0.047). The overall, mean number of sleep hours in a typical 24 h period is 7.37 h. Half of the participants were short sleepers (≤6 h/day), 41.6% slept for the recommended hours (7–9 h/day), while 7.3% were long sleepers (≥9 h/day). A statistically significant association was observed between sleep category and gender (*p* = 0.001)

### 3.3. Fruit and Vegetable Consumption

Number of Servings of Fruit and Vegetables Consumed on a Typical Day

Overall, the participants consumed less than two servings of fruit (mean servings 1.72) and slightly less than two (mean servings 1.91) servings of vegetables per day (Table 3). Fruit was consumed in more than three days (mean days 3.87) in a typical week, with a statistically significant gender difference (*p* = 0.002). Females consumed fruit in more days in a week in comparison with males (Table 3). On the other hand, vegetables were consumed on five days in a typical week, with a mean of 5.00. The analysis shows minimal differences in the mean number of days of vegetable consumption between males and females, with a mean of 4.98 days per week for males, and a mean of 5.03 days per week for females. Nearly half (47.0%) of the participants consumed vegetables every day of the week, and nearly one-third (31.4%) of them consumed vegetables on three to four days in a typical week. 

More than three-quarters of the participants (78.5%) consume inadequate amounts of fruit and vegetables per day (<5 servings), less than recommended by the WHO. More females (24.7%) than males (16.7%) consume five servings or more of fruit and/or vegetables on average per day. No statistically significant relationships were found between gender and consumption of the recommended number of servings of fruit and/or vegetables as stipulated by the WHO (*p* = 0.050). The results indicate that seven in ten participants were unaware of the recommendations for fruit and vegetable intake. However, of the 29.8% who reported to be aware of the recommendations, only 16.8% were able to indicate the number of servings of fruit and vegetables recommended by the WHO (at least five portions (400 g) of fruits and/or vegetables) per day. 

### 3.4. Prevalence of Combined Risk Factors for Chronic Diseases of Lifestyle

Chronic diseases of lifestyle (CDLs) have five common risk factors including: current daily smokers, consumption of fewer than five servings of fruit and vegetables per day, being overweight (BMI ≥ 25.0), low levels of physical activity (<600 METs), and raised blood pressure (SBP ≥ 140 and/or DBP ≥ 90 mmHg) [34]. Only 8.7% of the participants were classified as low risk for CDL (having none of the five risk factors), 48.9% had one risk factor, 32.9% had two risk factors, 8.5% had three risk factors, while the minority 0.9% had four risk factors (Table 4). 

### 3.5. Association between Fruit and Vegetable Intake and Overweight

No correlation was established between fruit and vegetable intake levels and being overweight (BMI ≥ 25.0) (*p* = 0.293). However, participants who consume fruit daily, or who consume vegetables daily, or who met the WHO recommendation of consuming at least five servings of fruit and/or vegetables per day were 30.0%, 48.0%, and 36.0% more likely to be overweight, respectively (Table 5). This could be attributed to the health consciousness effect. It could be that overweight individuals are consuming a healthy diet that incorporates adequate intake of fruit and vegetables to lose or prevent weight gain, as recommended by healthy eating guidelines that emphasize the importance of adequate intake of fruit and vegetables in the prevention of chronic diseases of lifestyle. Overweight in the study population could also be attributed to factors other than daily and adequate consumption of fruit and vegetables. 

No significant correlation was found between daily vegetable consumption or meeting the WHO recommendation of five or more servings of fruit and vegetables a day with waist circumference (*p* = 0.350). There was, however, a statistically significant relationship between daily fruit consumption and waist circumference, and daily fruit consumers were twice as likely to have a higher waist circumference than non-daily consumers (*p* = 0.006). In addition, no significant association was found between daily vegetable consumption or meeting of the WHO recommendation of five or more servings of fruit and vegetables a day with WHtR (*p* = 0.327). However, a significant relationship exists between daily fruit consumption and waist–height ratio (*p* = 0.013), and the participants consuming fruit daily were 89.0% more likely to have a high WHtR as non-daily fruit consumers. 

Findings of the binary logistic regression showed no significant relationship existed between daily fruit intake (*p* = 0.635), daily vegetable intake (*p* = 0.431), or consumption of five or more servings of fruit and vegetables per day (*p* = 0.573) and the waist–hip ratio.

### 3.6. Association between Fruit and Vegetable Intake and Blood Pressure, Blood Glucose and Total Cholesterol

Findings from correlations found no significant relationship between daily fruit consumption (*p* = 0.272), daily vegetable consumption (*p* = 0.805), or meeting the WHO recommendation of consuming five or more servings of fruit and vegetables per day and blood pressure (*p* = 0.513), random blood glucose (*p* = 0.359), and total cholesterol (*p* = 0.606). However, further analysis revealed that participants consuming vegetables daily were 10.0% less likely to have raised total cholesterol than non-daily vegetable consumers.

## 4. Discussion

Family history of diabetes and high cholesterol were significantly correlated with the participants’ BMI. The literature shows that family history of CDL can increase the likelihood of developing a CDL or its risk factors [35,36,37,38]. A survey among university students in Colombia reported a similar finding, with an association between a family background of excess weight and obesity [39]. The findings suggest that a family history of CDL in the sample could predispose them to a greater risk of developing CDL at an earlier stage of their life. This study’s findings suggest that the issue of family history of CDL among young adults requires greater attention, which could be achieved through health promotion programmes with a focus on the adoption of healthier lifestyles to limit their risk to CDL such as diabetes, cardiovascular diseases, and cancer, among others. 

Overweight and obesity in young adults are major public health concerns worldwide. The Kenya STEPwise 2015 survey indicated that 16.6% of young adults aged from 18 to 29 years were overweight and 4.9% were obese [2], this study revealed slightly lower rates, with 13.9% and 4.3% being overweight and obese, respectively. Other studies in younger adults show a higher prevalence than reported in this study. The prevalence of overweight and obesity was 20.4% and 14.9% in a Saudi Arabian university [40], 21.7% and 3.0% in three Cameroonian universities [41], 23.2% and 11.3% in a US university [42], 25.1% and 7.6% in a university in Colombia [39], and 27% and 8.5% in an Egyptian university, respectively [43]. 

In a South African university, more males were found to be more overweight (35.3%) than obese (6.0%), when in comparison, females had a lower prevalence of overweight (7.1%) and obesity (1.4%) [44]. A survey in another South African university showed a similar trend, where more males (41%) were overweight or obese in comparison with females (25%) [45]. This study had similar findings to the 2015 Kenya STEPwise survey conducted in the general population which reported that women were more likely to be overweight (24.9%) and obese (13.7%) than the men, with 13.2% of men being overweight and 4.3% obese [2]. Overweight and obesity in young adults pose a risk of development of CDL in early adulthood or later in life. The current findings show that overweight and obesity in young adults should be given more attention to further foster the programmes for the prevention of CDL such as type 2 diabetes, cardiovascular diseases, and cancers to enable prevention of CDL at an earlier stage. 

In the present study sample, the overall mean of total body fat percentage was higher among females (30.4%) than males (14.1%) and within the healthy range. However, the higher body fat percentage among females is attributed to women having a biologically higher percentage of body fat than men. These findings are in line with a study by Kęska et al. [46] conducted among university students who reported a higher mean of fat percentage among females (26.0%) compared with males (15.8%) [46]. Research in a Croatian university found that females had a higher mean fat percentage (31.8%) than males (22.3%) [26]. Similar findings were reported in Italy [47].

Current findings revealed a statistically significant difference between the WC and sex (*p* = 0.0001), where male participants had a higher mean waist circumference (74.1 cm) compared with females (73.4 cm), both within the healthy ranges. In contrast with these study findings, a higher WC mean was reported among males (83.4 cm) than females (75.9 cm) at a South African university [45] and among young male (89.7 cm) Peruvian students than female (86.1 cm) [48]. In the present study, females (25.9%) were more likely to be classified with central obesity compared with males (2.4%). The current finding is similar to research conducted in three universities in Cameroon, where 21.2% of the study sample had central obesity [41]. This similarity is most probably due to the similarities in target population and age between the studies.

The waist–height ratio is an anthropometric index used in the assessment of central obesity. Findings of the current study show females had a higher mean WHtR (0.45) compared with males (0.43), though both means are within the expected ratio of below 0.5. This result is in line with a study conducted at three universities in Bangladesh, which reported a higher mean among females (0.51) than males (0.48) [49], and Cameroon, which reported a higher mean among female students (0.51) in comparison with males (0.47) [50]. Contrary to the present study findings, the mean of WHtR was significantly higher among males (0.55) than females (0.54) in young Peruvian students [48]. This study established that 18.4% of the participants were centrally obese (WHtR ≥ 0.5), and comparatively, a statistically significant difference was found between sex and WHtR (*p* = 0.0001), with a higher prevalence reported among females (25.5%) than males (7.7%). In contrast with the current study findings, three studies among young adults in South Africa found that 71.4% [51], 44% [52] and 68% [53] were centrally obese. 

The present study established a significantly higher WHR mean among male participants (0.81) in comparison with females (0.75), all within the healthy ranges for gender. This finding is in line with the results of a study conducted among university students in Cameroon, where males had a higher mean (0.81) than females (0.79) [41]. Also, the study results showed that 10.7% of the males and 8.6% of the females had a higher waist–hip ratio; higher than the healthy ranges as classified by the WHO. These findings were not in line with findings of a Kenyan survey that found a higher WHR among females (28.4%) compared with males (17.5%) [2]. In contrast to the above studies, more male Ghanaian students (55.9%) were found to have a higher WHR than female students (27.3%) [54]. 

In this study, a significantly higher systolic blood pressure was observed among males (123 mmHg) than among females (115 mmHg) (*p* = 0.0001), and there is a higher mean DBP for males than females (75.1 mmHg), with a p-value of 0.001. This is consistent with studies conducted among young adults in South Africa, where a study showed that male participants had a higher SBP (125.5 mmHg) than females (109.1 mmHg), and DBP means were 73.34 mmHg among males compared with 71.86 mmHg in females [45]. 

Eight percent of the participants in the present study were diagnosed with raised blood pressure (SBP ≥ 140 and/or DBP ≥ 90 mmHg), with more male participants being diagnosed with raised blood pressure (13.7%) in comparison with females (4.7%), *p* = 0.002. The current finding is slightly lower than the 2015 Kenya STEPS survey that revealed 13.2% of young adults had raised blood pressure and that it was more prevalent among males (15.4%) than females (11.1%) [2]. However, these findings were inconsistent with young adults in Cameroon, where 6.3% of young adults had raised blood pressure, with more males (7.0%) compared with females (4.7%) [55], and a Zambian study found a higher prevalence of raised blood pressure among females (3.1%) in an urban population than males [56].

In this study, the mean random blood glucose for all participants was 4.8 mmol/L, with a slightly higher mean among females (4.9 mmol/L) than males (4.8 mmol/L), though this is not statistically significant. University students in a South African study recorded an overall mean of 5.67 mmol/L, where males were found to have a significantly higher mean random blood glucose value (5.82 ± 1.03 mmol/L) compared with females (5.61 ± 0.85 mmol/L) (*p* = 0.02) [45]. Borderline raised blood glucose (7.8–11.1 mmol/L) was recorded in 0.5% of the study sample. This prevalence is consistent with data from the 2015 Kenya STEPwise survey, which reported that 0.5% of young adults had raised blood glucose [2]. 

In the current study, mean total cholesterol for all participants was 4.7 mmol/L, with a significant difference (*p* = 0.0001) between the mean of males (4.9 mmol/L) and females (4.6 mmol/L). This study’s overall mean was similar to the overall mean (4.61 mmol/L) reported in university students in South Africa; however, female participants had a significantly higher total weighted mean cholesterol (4.67 mmol/L) compared with males (4.47 mmol/L) (*p* = 0.0001) in the South African study [45]. The current findings are also not in line with the mean (3.4 mmol/L) established among young adults in the Kenyan survey, with a higher mean among females (3.6 mmol/L) than males (3.2 mmol/L) [2]. 

In the present study, 39.5% of the study sample had raised total cholesterol (≥5.0 mmol/L). A significantly higher proportion of males (51.2%) had raised cholesterol compared with females (31.8%), with a p-value of 0.0001. These findings are inconsistent with the nationally representative survey in Kenya that showed 5.6% of young adults aged from 18 to 29 years had raised total cholesterol, with a higher prevalence reported among females (8.8%) than males (2.1%) [2]. The findings show 4.0% of the participants had high total cholesterol (≥6.2 mmol/L), with a significantly higher prevalence among male participants (8.9%) compared with female participants (0.8%). This is in contrast with the Kenya STEPwise survey that established that only 0.6% of the young adults had high total cholesterol, with more women (1.1%) than men (0.1%) [2].

The current findings on current smokers were in line with the findings of a study conducted among university students from 28 LMICs and high-income countries, which found that 11.8% of the study population were current tobacco users [57]. Other studies on young adults conducted in Australia, the US, and Croatia revealed a higher prevalence of current smokers, where 22.7% [7], 24.6% [58] and 57.2% [26], respectively, were smokers. More males (66.0%) than females (58.9%) were reported in a Croatian study [26]. A lower prevalence was also reported among Cambodian university students where 1.1% were found to be current smokers [59]. 

Ten percent (10.7%) of males and 4.7% of female students reported harmful alcohol consumption in this study sample. This is inconsistent with the Kenya STEPwise survey report, where one in ten (9.9%) young adults reported being heavy episodic drinkers (≥6 drinks). More males (17.7%) than females (2.8%) were heavy drinkers [2]. The difference is attributed to the classification of heavy episodic drinkers used in the study (≥6 drinks for both genders) compared with the current study classification (≥5 drinks in males, ≥4 drinks in females). Consistent with the present study, a survey conducted among university students in 28 countries, reported that 10.6% of the sample reported harmful alcohol consumption [57]. Harmful consumption was observed among 40% of male participants and 28% of female students in a South African study [60]. 

Current findings showed 36.2% of the sample had lower physical activity, 41.4% were moderately active, whilst 22.5% had a high level of physical activity. More female participants (43.1%) had lower physical activity levels compared with males (25.6%), with a significant difference (*p* = 0.0001). This differs to the Kenya STEPwise survey, where 71.1% (74.4% males vs. 67.9% females) of young adults (18–29 years) had a high level of physical activity, 16.4% (12.9% males, 19.8% females) were moderately active, and 12.5% (12.7% males, 12.3% females) had a low level of total physical activity [2]. The difference is attributed to the present study classification of the levels of physical activity based on a combination of MET minutes of moderate- and vigorous-intensity activity in a week. In line with the current study, studies in young adults found that more females were physically inactive than males. 

Sleep deprivation is related to unhealthy food consumption and predisposes young adults to the risk of developing NCDs [61,62]. In this study, the average sleeping duration was 7.37 h on a typical day. This is in line with a study conducted on young adults in a Cambodian university which reported a mean of 7.2 h of sleep [59]. Current findings showed that 51.1% of the sample had inadequate sleep duration whilst 41.6% reported to have adequate sleep duration. This was inconsistent with a study by Pengpid et al., (2020) among university students from 28 low, middle, and high income countries, which found that 38.9% of the sample had inadequate sleep [57]. One-third and one-quarter of young adults reported inadequate sleep in studies conducted in the US and South Africa, respectively [45,63]. A smaller proportion (4.2%) of students in thirteen Dutch universities reported inadequate sleep [64]. 

This study established that only one-fifth (21.5%) of the participants consumed five or more servings of fruit and/or vegetables per day. More females (24.7%) than males (16.7%) consume five or more servings of fruit and/or vegetables on average per day. No statistically significant relationships were found between sex and consumption of adequate servings of fruit and/or vegetables as recommended by the WHO (*p* = 0.050). These findings were consistent with recent studies conducted on young adults. Results from a survey conducted among university students in 28 countries across Asia, Africa and America, found that only 17.2% consumed five portions of fruit and/or vegetables per day [65]. Other studies conducted among university students in Egypt, Saudi Arabia, Singapore, the United States of America, the United Kingdom, and New Zealand have reported low fruit and vegetable intake and fewer young adults are meeting the recommended level of five combined servings of fruits and vegetables per day [6,24,42,43,66,67,68]. The current findings depict a similar picture revealed by a recent WHO-STEPS survey in African countries, which showed very low proportions of young adults aged 18 to 29 years consumed five or more servings of fruits and/or vegetables per day [69,70,71,72,73,74,75,76,77,78].

### Knowledge of Fruit and Vegetable Intake

Nutrition knowledge is a component of health literacy, and a potential predictor of dietary intake and patterns, including fruit and vegetable intake [79]. The current study findings are in line with a study conducted on young adults in Egypt, that showed that a smaller proportion (8.2%) of the study population were aware of the recommendation to consume five daily servings of fruit and vegetables [43]. In contrast to this study, a study in the UK where the majority of the sample were aware of the recommendation for daily consumption of fruit and/or vegetables [80]. 

Current findings further revealed that only 16.8% of those who reported to be aware of the recommended fruit and vegetables intake were aware of the adequate number of servings of fruit and vegetables as recommended by the WHO of at least (400 g) five portions of fruits and/or vegetables per day. This is consistent with findings of studies in the UK and Australia, which found that even though the participants were aware of the fruit and vegetable intake guidelines, they lacked knowledge on portion sizes and the need for variety concerning fruit and vegetable consumption [80,81]. 

The results of this study have several important policy implications for health system strengthening in the context of CDL. They identify the dire need to improve the consumption of fruits and vegetables and prevent CDL in young adults. Policies need to focus on a nationally integrated long-term campaign to increase fruit and vegetable consumption and sensitization to the risk factors of CDL among young adults, supported by policy changes to address structural barriers to consumption and accessibility to healthcare. Although there is no sufficient evidence to support sex-specific interventions and policies, future public health initiatives should consider these differences. Our results showed significant sex differences in fruit and vegetable consumption patterns and prevalence of CDL risk factors. Future studies should investigate barriers to adequate fruit and vegetable intake among young adults. Further longitudinal research is needed to explore the modifiable determinants of fruit and vegetable consumption changes in early adulthood and understand differences in dietary trajectories among different population groups. This will provide evidence for targeting public health policy.

The main limitation of the study was that it was a cross-sectional study. This limits the drawing of conclusions about causal associations. Lack of association between fruit and vegetable intake and the risk factors of CDL could be attributed to other factors beyond consumption of sufficient amounts of fruit and vegetables such as biological and behavioural risk factors. Volunteer participation in the study could be subject to voluntary bias, which limits data generalisability to the general population. The study utilised data from a survey among young adults in a Kenyan university, limiting the generalisation of the results to the general population and Africa. Other potential limitations include that the participants’ responses are subject to recall bias and response bias such as social desirability common in self-administered questionnaires. However, despite these potential limitations, to the best of our knowledge, this is the first study in Kenya and the region to assess fruit and vegetable consumption patterns and CDL risk in young university adults. This study contributes to the literature on fruit and vegetable consumption patterns and the prevalence of risk factors for chronic diseases of lifestyle in young adults in Kenya and Africa. As our sample was representative of all regions in Kenya it adds to the richness and depth of the data. The data obtained during this cross-sectional study will further knowledge of fruit and vegetable consumption patterns and the risk factors of CDL in this population. 

## 5. Conclusions

The findings of this study highlight a shortfall of fruit and vegetable consumption in young adults. More than four-fifths of the study sample consumed lower amounts of fruit and vegetables than recommended by the national guidelines and the WHO. Gender differences regarding the intake of fruit and vegetables are notable, although non-significant statistically. Biological risk factors were observed in this study. More than half, nearly one-third, and nearly one-fifth of the study sample had a family history of high blood pressure, diabetes, and high cholesterol, respectively. The study revealed that nearly one-fifth of the study sample were overweight or obese and had a higher waist–height ratio, one-quarter had a higher total body fat percentage, sixteen in a hundred participants had a higher waist circumference, more than one-third had raised total cholesterol, and 8.3% had raised blood pressure. 

Overall, the risk factors of chronic diseases of lifestyle are prevalent in the study sample, with 91% having at least one combined risk factor of chronic diseases of lifestyle. The department of non-communicable diseases should give particular attention to fruit and vegetable consumption and other risk factors of chronic diseases of lifestyle prevalent in young adults. The fact that these risk factors for chronic diseases of lifestyle have health implications later in life prompts the need to strengthen public health nutrition strategies to increase fruit and vegetable consumption in young adulthood and reduce chronic diseases of lifestyle later in life.

## Figures and Tables

**Table 1 ijerph-19-06965-t001:** Demographic and socioeconomic characteristics of the study participants.

Variable	Frequency (n)	Percentage (%)
Gender		
Male	168	39.7
Female	255	60.3
Age		
19–24	337	79.7
25–30	86	20.3
Race		
African	423	100.0
Religion		
Christian	404	95.5
Islam	17	4.0
Hinduism	2	0.5
Faculty enrolled *		
Arts, Humanities, and Social Sciences	80	18.9
Business and Economics	23	5.4
Education	166	39.2
Engineering and Technology	19	4.5
Hospitality and Tourism Management	15	3.5
Medicine and Health Sciences	108	25.5
Pure and Applied Sciences	12	2.8
Region in Kenya		
Central Region	69	16.3
Coastal Region	23	5.4
Eastern Region	65	15.4
Nairobi Region	80	18.9
Rift Valley Region	84	19.9
North-Eastern Region	2	0.5
Nyanza Region	74	17.5
Western Region	26	6.1
Income (those who work)		
Yes	61	14.5
No	362	85.5
Allowance (sponsors, parents or grants)		
Yes	82	19.5
No	341	80.5
Residency		
Living with family	144	34.0
Living alone	145	34.3
Shared/university accommodation	134	31.7
Have an orchard at home		
Yes	114	27.0
No	309	73.0
Have a vegetable garden		
Yes	225	53.2
No	198	46.8
Planning of a typical day food menu		
Family member/cook	186	44.0
Self	237	56.0
Total number of family members		
0–5 Members	252	59.6
6–10 Members	152	35.9
More than 10 Members	19	4.5
Total household monthly income *		
Below KES 20,000	75	17.8
KES 20,001–40,000	56	13.2
KES 40,001–100,000	50	11.8
Above KES 100,001	38	8.9
Unknown	204	48.2

* For purposes of reporting the participants schools were grouped into 7 faculties from 19 schools. KES is Kenyan Shilling where 1 USD = KES 101.15 at the time of data collection in 2019.

**Table 2 ijerph-19-06965-t002:** Mean of participants physical measurements.

Mean of Physical Measurements				
Measurement	Male		Female	
	Mean	95% CI	Mean	95% CI
Height (cm)	173.3	172.2–174.5	162.2	161.4–163.0
Weight (kg)	62.9	61.4–64.3	60.3	58.8–61.9
BMI (kg/m^2^)	20.9	20.5–21.4	22.9	22.5–23.5
Total body fat (%)	13.4	12.7–14.4	28.5	27.4–29.5
Visceral fat (%)	2.1	1.8–2.3	2.4	2.2–2.7
Muscle mass (kg)	51.4	50.4–52.2	40.2	39.7–40.6
Bone density (kg)	2.7	2.7–2.8	2.2	2.1–2.2
Basal metabolic rate (Kcal)	1603.2	1576.2–1629.1	1329.5	1314.5–1344.6
Total body water (%)	61.3	60.6–62.1	50.1	49.4–50.7
Waist circumference (cm)	74.1	72.9–75.3	73.4	72.1–74.7
Waist/height ratio (WHtR)	0.43	0.42–0.44	0.45	0.44–0.46
Waist-to-hip ratio	0.81	0.80–0.82	0.75	0.74–0.76
Systolic blood pressure (mmHg)	123.5	121.8–125.6	115.0	113.7–116.4
Diastolic blood pressure (mmHg)	77.9	76.6–79.4	75.1	74.2–76.1

**Table 3 ijerph-19-06965-t003:** Mean number of days and servings of fruit and vegetables consumed in a typical day and week.

Mean Number of Days and Servings of Fruit and Vegetables Consumed in a Typical Day and Week										
Variable	Males			Females			All			*p*-Value
	n	Mean	95% CI	n	Mean	95% CI	n	Mean	95% CI	
Mean number of days fruit is consumed in a week.	168	3.44	3.03–3.86	255	4.16	3.89–4.48	423	3.87	3.63–4.10	0.002
Mean number of days vegetables are consumed in a week.	168	4.98	4.64–5.27	255	5.03	4.77–5.32	423	5.00	4.78–5.22	0.682
Mean number of servings of fruit a day.	168	1.65	1.51–1.81	255	1.77	1.64–1.90	423	1.72	1.62–1.82	0.237
Mean number of servings of vegetables a day.	168	1.89	1.73–2.06	255	1.93	1.81–2.06	423	1.91	1.82–2.01	0.710
Mean number of servings of fruit and/or vegetables a day.	168	3.54	3.31–3.80	255	3.70	3.50–3.91	423	3.64	3.49–3.79	0.341

**Table 4 ijerph-19-06965-t004:** Percentage of participants who showed risk factors of chronic diseases of lifestyle.

Risk Factor	Cut-Off Point	% Males	% Females	% All
**Behavioural Risk Factors**				
Smoking	Daily smoker	6.0	0.8	2.8
Alcohol consumption	Males ≥ 6 drinks Females ≥ 4 drinks	8.3	4.7	6.1
Fruit and vegetable consumption	<5 servings of fruit and vegetables.	83.3	75.3	78.5
Physical activity	<600 MET minutes	25.6	43.1	36.2
**Biological Risk Factors–Physical and Biochemical Measurements**				
Body mass index	BMI ≥ 25.0	7.7	25.1	18.2
Total body fat percentage	Males > 23% Females > 31%	14.3	32.9	25.5
Waist circumference	Males ≥ 94.0 cm Females ≥ 80 cm	2.4	25.9	16.5
Waist–hip ratio	Males ≥ 0.90 Females ≥ 0.85	10.7	8.6	9.5
Waist–height ratio	≥0.5	7.7	25.5	18.4
Blood pressure	(SBP ≥ 140 and/or DBP ≥ 90 mmHg)	13.7	4.7	8.3
Random blood glucose	≥7.8 mmol/L	0.6	0.4	0.5
Total cholesterol	≥5.0 mmol/L	51.2	31.8	39.5
**Biological Risk Factors—Family History of Chronic Diseases of Lifestyle**				
Blood pressure		45.8	58.0	53.2
Diabetes		28.0	34.5	31.9
Raised cholesterol		16.1	18.0	17.3
Cancer		16.7	13.7	14.9
Stroke		8.3	12.9	11.1
Heart disease		7.7	7.1	7.3

**Table 5 ijerph-19-06965-t005:** Association of fruit and vegetable consumption with biological risk factors.

Variables	Fruit and Vegetable Consumption		*p*-Value	Odds Ratio (95% CI)
	<5 Servings of Fruit and Vegetables n (%)	≥5 Servings of Fruit and Vegetables n (%)		
BMI ≥ 25.0 Normal weight Overweight	275 (82.8) 71 (78.0)	57 (17.2) 20 (22.0)	0.293	1.36 (0.77–2.41)
Total body fat percentage Normal percentage Overweight	248 (74.7) 67 (73.6)	84 (25.3) 24 (26.4)	0.835′	1.06 (0.62–1.79)
Waist circumference Normal Central Obesity	280 (84.3) 73 (80.2)	52 (15.7) 18 (19.8)	0.350	1.33 (0.73–2.41)
Waist to height ratio Normal Central Obesity	274 (82.5) 71 (78.0)	58 (17.5) 20 (22.0)	0.327	1.33 (0.75–2.36)
Waist to hip ratio Normal Central Obesity	302 (91.0) 81 (89.0)	30 (9.0) 10 (11.0)	0.573	1.24 (0.58–2.65)
Blood pressure Normal Raised BP	303 (91.3) 85 (93.4)	29 (8.7) 6 (6.6)	0.513	0.74 (0.30–1.84)
Random blood glucose Normal Raised Blood Glucose	331 (99.7) 90 (98.9)	1 (0.3) 1 (1.1)	0.359	3.68 (0.23–59.38)
Total cholesterol Normal Raised Cholesterol	203 (61.1) 53 (58.2)	129 (38.9) 38 (41.8)	0.616	1.13 (0.70–1.81)

## Data Availability

The data presented in this study are available on request from the corresponding author. The data are not publicly available due to using databases that are not online.

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
