# Peer review of "Fruit and Vegetable Consumption Patterns and Risk of Chronic Diseases of Lifestyle among University Students in Kenya"

_ijerph, 2022, doi:10.3390/ijerph19126965_

Round 1
Reviewer 1 Report
I suggest the authors remove from the text "Error! Reference source not found", possibly references may be made where they are given to other similar studies in other countries. In fact, it is interesting to make a much more complex correlation of data with similar studies from other countries.
Author Response
th May 2022
Dear Editor,
RE: RESPONSES TO REVIEWERS COMMENTS/SUGGESTIONS
Please find enclosed our revised manuscript. We thank the four reviewers for their constructive comments on our manuscript. We have fully addressed the comments in the revised the manuscript and we believe these revisions have resulted in a significantly improved manuscript.
Reviewer 1:
- I suggest the authors remove from the text "Error! Reference source not found",
- “The error! Reference source not found” maybe appearing due the version mode used to save the manuscript. We have changed into compatibility mode to avoid this from happening.
- Possibly references may be made where they are given to other similar studies in other countries. In fact, it is interesting to make a much more complex correlation of data with similar studies from other countries.
- We have made comparisons of our results with studies in young adults in other countries.

Reviewer 2 Report
Dear authors. This is a very interesting article on a very important topic. We would like to make some corrections and suggestions:
- Present fewer keywords in the abstracts. Keywords should not be the same as in the title.
- Be consistent in reporting results, especially percentages (e.g. in the abstract, report both with one decimal 75.0% and without the decimal 85%).
- Use fewer references per sentence (e.g. 10-18 references per sentence).
- When you first introduce a word or phrase with an abbreviation, use it again in the future. Do not use the whole word and the abbreviation interchange.
- In the methodology, add why you used certain parameters in the sample calculation.
- Please add in the methodology whether these are calibrated instruments/validated questionnaires.
- Specify the p-value in the methodology.
- In the methodology, it says that the research will take place at the University of Kenya. You cannot generalise the data to the whole of Africa, which is also a limitation.
- We recommend using the average age.
- We recommend that the results are consistent in the text.
- Some online references do not exist.
Author Response
Reviewer 2:
- Present fewer keywords in the abstracts. Keywords should not be the same as in the title.
- We have made changes. Reduced the number of keywords from nine to six keywords.
- Be consistent in reporting results, especially percentages (e.g. in the abstract, report both with one decimal 75.0% and without the decimal 85%).
- We have made changes in the abstract, and presentations of results in the results and discussion section. Using one decimal place for consistency.
- Use fewer references per sentence (e.g. 10-18 references per sentence).
- The suggestion have been implemented.
- When you first introduce a word or phrase with an abbreviation, use it again in the future. Do not use the whole word and the abbreviation interchange.
- We have made changes to improve consistency. I have used the phrases and abbreviations when I first introduced CDL, WHR, BMI, WHO among other abbreviations.
- In the methodology, add why you used certain parameters in the sample calculation.
- The suggestion have been implemented.
- Please add in the methodology whether these are calibrated instruments/validated questionnaires.
- We have made suggested changes in the methodology section. The survey adopted validated questionnaire from the WHO STEPs survey. The assessment instruments were calibrated before data collection commenced, daily.
- Specify the p-value in the methodology.
- We have made changes and incorporated them in the data analysis section.
- In the methodology, it says that the research will take place at the University of Kenya. You cannot generalise the data to the whole of Africa, which is also a limitation.
- We have added this limitation in the limitations section.
- We recommend using the average age.
- We have highlighted the mean age of the participants before table 1.
- We recommend that the results are consistent in the text.
o The suggestion have been implemented.
- Some online references do not exist.
We have reviewed all references and included new links to the references affected.
Reviewer 2:
- Present fewer keywords in the abstracts. Keywords should not be the same as in the title.
- We have made changes. Reduced the number of keywords from nine to six keywords.
- Be consistent in reporting results, especially percentages (e.g. in the abstract, report both with one decimal 75.0% and without the decimal 85%).
- We have made changes in the abstract, and presentations of results in the results and discussion section. Using one decimal place for consistency.
- Use fewer references per sentence (e.g. 10-18 references per sentence).
- The suggestion have been implemented.
- When you first introduce a word or phrase with an abbreviation, use it again in the future. Do not use the whole word and the abbreviation interchange.
- We have made changes to improve consistency. I have used the phrases and abbreviations when I first introduced CDL, WHR, BMI, WHO among other abbreviations.
- In the methodology, add why you used certain parameters in the sample calculation.
- The suggestion have been implemented.
- Please add in the methodology whether these are calibrated instruments/validated questionnaires.
- We have made suggested changes in the methodology section. The survey adopted validated questionnaire from the WHO STEPs survey. The assessment instruments were calibrated before data collection commenced, daily.
- Specify the p-value in the methodology.
- We have made changes and incorporated them in the data analysis section.
- In the methodology, it says that the research will take place at the University of Kenya. You cannot generalise the data to the whole of Africa, which is also a limitation.
- We have added this limitation in the limitations section.
- We recommend using the average age.
- We have highlighted the mean age of the participants before table 1.
- We recommend that the results are consistent in the text.
o The suggestion have been implemented.
- Some online references do not exist.
We have reviewed all references and included new links to the references affected.
Reviewer 3 Report
Many thanks for the opportunity to review this interesting manuscript. The manuscript focuses on fruit and vegetable consumption patterns and risk factors for chronic diseases of lifestyle among young adults attending Kenyatta University in Kenya. The evidence showed a shortfall in fruit and vegetable consumption in young adults. However, there were some unclear parts, particularly in the Methods section. Therefore, the results are not fully reproducible based on the details given in the Methods section.
Please find a couple of comments below.
- Please consider stating that this is a cross-sectional study in the title or abstract.
- Line 28: You state, ‘’ In the past few decades, chronic diseases of lifestyle (CDLs) such as cardiovascular diseases, cancers, type 2 diabetes, and obesity are affecting younger populations at an alarmingly higher rate than before’’. Please report some numbers or examples.
- Line: 74, Methods: How the study participants were recruited? By whom?
- Line: 74, Methods: Where were these measurements done?
- Line: 74, Methods: What is the recruitment period?
- Line: 74, Methods: What is the data collection period?
- Line 102: Who did the sampling and screening?
- Line 103: What were the inclusion and exclusion criteria?
- Lines 102 to 109: Unclear; please clarify.
- Line 115: You mention behavioral risk factors, biological risk factors, and physical measurements; what are these specifically?
- Line 141: Who did the biomedical assessments? Were there any protocols?
- Line 151: Please describe all the statistical analyses used.
- Line 151: Were there any missing data variables?
- Line 218: The Methods section does not mention tobacco use or physical activity.
- Could you please report the number of individuals at each stage of the study? For instance, what was the number of the potentially eligible population? How many of them participated? How many of the participants were finally analyzed?
- Many ‘’Error! Reference source not found.’’ warnings in the Results section.
- There are some repetitions of results in the Discussion section. Could you please provide some broader discussion on the implications?
Author Response
Reviewer 3:
- Please consider stating that this is a cross-sectional study in the title or abstract.
- We have highlighted that our study was a cross-sectional study in the abstract.
- Line 28: You state, ‘’ In the past few decades, chronic diseases of lifestyle (CDLs) such as cardiovascular diseases, cancers, type 2 diabetes, and obesity are affecting younger populations at an alarmingly higher rate than before’’. Please report some numbers or examples.
- We have made changes and included projected numbers from the WHO.
- Line: 74, Methods: How the study participants were recruited? By whom?
- We have made changes. We have highlighted the steps in recruitment and personnel involved in the sampling section.
- Line: 74, Methods: Where were these measurements done?
- We have made changes. We have included the places in the data collection section.
- Line: 74, Methods: What is the recruitment period?
- We have made changes and included the period in the sampling section.
- Line: 74, Methods: What is the data collection period?
- We have made changes. We have included the period in the data collection section.
- Line 102: Who did the sampling and screening?
- We have made changes and incorporated them in the sampling and data collection sections.
- Line 103: What were the inclusion and exclusion criteria?
- We have made changes. We have included the inclusion and exclusion criteria under the study population.
- Lines 102 to 109: Unclear; please clarify.
- We have made changes in the sampling section.
- Line 115: You mention behavioral risk factors, biological risk factors, and physical measurements; what are these specifically?
- We have made changes and included more details in data collection section.
- Line 141: Who did the biomedical assessments? Were there any protocols?
- We have made changes. We have highlighted it under data collection section under biochemical assessments. We utilized a priori standards of procedures protocol.
- Line 151: Please describe all the statistical analyses used.
- We have made changes. We have highlighted the statistical analyses used in the data analysis section.
- Line 151: Were there any missing data variables?
- We have included the information in the sampling section.
- Line 218: The Methods section does not mention tobacco use or physical activity.
- We have included more information in the data collection section.
- Could you please report the number of individuals at each stage of the study? For instance, what was the number of the potentially eligible population? How many of them participated? How many of the participants were finally analyzed?
- We have made changes and included in the sampling section.
- Many ‘’Error! Reference source not found.’’ warnings in the Results section.
- “The error! Reference source not found” maybe appearing due the Ms Word version mode used to save the manuscript. We have changed into compatibility mode to avoid this from happening.
- There are some repetitions of results in the Discussion section. Could you please provide some broader discussion on the implications?
- We have incorporated the implications to policy and research in the discussion section.
Reviewer 3:
- Please consider stating that this is a cross-sectional study in the title or abstract.
- We have highlighted that our study was a cross-sectional study in the abstract.
- Line 28: You state, ‘’ In the past few decades, chronic diseases of lifestyle (CDLs) such as cardiovascular diseases, cancers, type 2 diabetes, and obesity are affecting younger populations at an alarmingly higher rate than before’’. Please report some numbers or examples.
- We have made changes and included projected numbers from the WHO.
- Line: 74, Methods: How the study participants were recruited? By whom?
- We have made changes. We have highlighted the steps in recruitment and personnel involved in the sampling section.
- Line: 74, Methods: Where were these measurements done?
- We have made changes. We have included the places in the data collection section.
- Line: 74, Methods: What is the recruitment period?
- We have made changes and included the period in the sampling section.
- Line: 74, Methods: What is the data collection period?
- We have made changes. We have included the period in the data collection section.
- Line 102: Who did the sampling and screening?
- We have made changes and incorporated them in the sampling and data collection sections.
- Line 103: What were the inclusion and exclusion criteria?
- We have made changes. We have included the inclusion and exclusion criteria under the study population.
- Lines 102 to 109: Unclear; please clarify.
- We have made changes in the sampling section.
- Line 115: You mention behavioral risk factors, biological risk factors, and physical measurements; what are these specifically?
- We have made changes and included more details in data collection section.
- Line 141: Who did the biomedical assessments? Were there any protocols?
- We have made changes. We have highlighted it under data collection section under biochemical assessments. We utilized a priori standards of procedures protocol.
- Line 151: Please describe all the statistical analyses used.
- We have made changes. We have highlighted the statistical analyses used in the data analysis section.
- Line 151: Were there any missing data variables?
- We have included the information in the sampling section.
- Line 218: The Methods section does not mention tobacco use or physical activity.
- We have included more information in the data collection section.
- Could you please report the number of individuals at each stage of the study? For instance, what was the number of the potentially eligible population? How many of them participated? How many of the participants were finally analyzed?
- We have made changes and included in the sampling section.
- Many ‘’Error! Reference source not found.’’ warnings in the Results section.
- “The error! Reference source not found” maybe appearing due the Ms Word version mode used to save the manuscript. We have changed into compatibility mode to avoid this from happening.
- There are some repetitions of results in the Discussion section. Could you please provide some broader discussion on the implications?
- We have incorporated the implications to policy and research in the discussion section.
Reviewer 3:
- Please consider stating that this is a cross-sectional study in the title or abstract.
- We have highlighted that our study was a cross-sectional study in the abstract.
- Line 28: You state, ‘’ In the past few decades, chronic diseases of lifestyle (CDLs) such as cardiovascular diseases, cancers, type 2 diabetes, and obesity are affecting younger populations at an alarmingly higher rate than before’’. Please report some numbers or examples.
- We have made changes and included projected numbers from the WHO.
- Line: 74, Methods: How the study participants were recruited? By whom?
- We have made changes. We have highlighted the steps in recruitment and personnel involved in the sampling section.
- Line: 74, Methods: Where were these measurements done?
- We have made changes. We have included the places in the data collection section.
- Line: 74, Methods: What is the recruitment period?
- We have made changes and included the period in the sampling section.
- Line: 74, Methods: What is the data collection period?
- We have made changes. We have included the period in the data collection section.
- Line 102: Who did the sampling and screening?
- We have made changes and incorporated them in the sampling and data collection sections.
- Line 103: What were the inclusion and exclusion criteria?
- We have made changes. We have included the inclusion and exclusion criteria under the study population.
- Lines 102 to 109: Unclear; please clarify.
- We have made changes in the sampling section.
- Line 115: You mention behavioral risk factors, biological risk factors, and physical measurements; what are these specifically?
- We have made changes and included more details in data collection section.
- Line 141: Who did the biomedical assessments? Were there any protocols?
- We have made changes. We have highlighted it under data collection section under biochemical assessments. We utilized a priori standards of procedures protocol.
- Line 151: Please describe all the statistical analyses used.
- We have made changes. We have highlighted the statistical analyses used in the data analysis section.
- Line 151: Were there any missing data variables?
- We have included the information in the sampling section.
- Line 218: The Methods section does not mention tobacco use or physical activity.
- We have included more information in the data collection section.
- Could you please report the number of individuals at each stage of the study? For instance, what was the number of the potentially eligible population? How many of them participated? How many of the participants were finally analyzed?
- We have made changes and included in the sampling section.
- Many ‘’Error! Reference source not found.’’ warnings in the Results section.
- “The error! Reference source not found” maybe appearing due the Ms Word version mode used to save the manuscript. We have changed into compatibility mode to avoid this from happening.
- There are some repetitions of results in the Discussion section. Could you please provide some broader discussion on the implications?
- We have incorporated the implications to policy and research in the discussion section.
Reviewer 4 Report
This cross-sectional analytic study aims to assess the fruit and vegetable consumption patterns and lifestyle-related risk of chronic diseases among University students in Kenya. Authors assert a need for this study due to the fact that while several studies worldwide have assessed fruit and vegetable consumption among youth and shown young adults are not meeting recommended intake requirements, little is known about fruit and vegetable consumption patterns among young adults in Kenya and the rest of Africa. Authors aim to address this gap in the literature with their study.
The review is as follows:
- The acronym ‘CD’ (chronic diseases of lifestyle) should be spelled out the first time it is introduced.
- Line 28-32 – Specify if this information on the epidemiologic transition applies to youth in Kenya or youth worldwide, since the supporting references focus on Kenya.
- Lines 35-39 – This is insightful information. It would be interesting to know what contributes to this dietary shift in Kenya.
- Regarding the study population, how were participants recruited? Were incentives provided to participate in the study?
- There is insightful discussion of the sampling strategy.
- Line 104 – In “Specifically, those who met the inclusion criteria”, what was the inclusion criteria? Were there exclusion criteria?
- Lines 119-120 – Check wording and punctuation (consider comma after ‘needed’) in “Where needed the researcher and assistants was available to assist the participants”.
- Line 136 – The acronym ‘WHR’ should be written in full the first time it is introduced.
- Lines 165-166 – This sentence is unclear – “The socio-demographic characteristics of the study participants are presented in Error! Reference source not found..”. It is not clear what this statement is referring to.
- Lines 180-181 – Again this sentence is unclear – “The overall, mean total body fat percentage for males was 14.1% and 30.4% for females Error! Reference source not found..”.
- Lines 196 and 198; 210-211; 258-269; 261-262, 294 – The statement “Error! Reference source not found” is presented frequently and is unclear.
- Lines 239-240 – In “Overall, 52 minutes was spent on a typical day on recreation-related activity. Male participants spend more time (58.7 minutes) compared to females (42.5 minutes) per day”, authors should provide the recommended recreation-related activity levels for reference.
Overall, it is pleasing to see this article on a pertinent, unique, and important topic. It is interesting to read. Attending to some clarifying questions about the methodology and study results may have to improve the paper. Authors should also address the frequent mention of “Error! Reference source not found”.
Author Response
Reviewer 4:
- The acronym ‘CD’ (chronic diseases of lifestyle) should be spelled out the first time it is introduced.
- We have made changes to improve consistency. I have used the phrases and abbreviations when I first introduced CDL, WHR, BMI, WHO among other abbreviations.
- Line 28-32 – Specify if this information on the epidemiologic transition applies to youth in Kenya or youth worldwide, since the supporting references focus on Kenya.
- Citations from the 2 MoH reports and 1 from the WHO highlighting the issue in Kenya.
- Lines 35-39 – This is insightful information. It would be interesting to know what contributes to this dietary shift in Kenya.
- Suggestion has been implemented.
- Regarding the study population, how were participants recruited? Were incentives provided to participate in the study?
- We have made changes. We have highlighted the steps in recruitment in the sampling section. Incentives were also provided and has been incorporated in the manuscript in the data collection section.
- There is insightful discussion of the sampling strategy.
- Line 104 – In “Specifically, those who met the inclusion criteria”, what was the inclusion criteria? Were there exclusion criteria?
- We have included the inclusion and exclusion criteria under the study population.
- Lines 119-120 – Check wording and punctuation (consider comma after ‘needed’) in “Where needed the researcher and assistants was available to assist the participants”.
- We have made changes. We have inserted a comma after the word “needed”.
- Line 136 – The acronym ‘WHR’ should be written in full the first time it is introduced.
- We have made changes. The full name has been inserted where the acronym is first introduced throughout the manuscript.
- Lines 165-166 – This sentence is unclear – “The socio-demographic characteristics of the study participants are presented in Error! Reference source not found..”. It is not clear what this statement is referring to.
- “The error! Reference source not found” maybe appearing due the Ms Word version mode used to save the manuscript. We have changed into compatibility mode to avoid this from happening. The statement was referring to table 1.
- Lines 180-181 – Again this sentence is unclear – “The overall, mean total body fat percentage for males was 14.1% and 30.4% for females Error! Reference source not found..”.
- “The error! Reference source not found” maybe appearing due the Ms Word version mode used to save the manuscript. We have changed into compatibility mode to avoid this from happening.
- Lines 196 and 198; 210-211; 258-269; 261-262, 294 – The statement “Error! Reference source not found” is presented frequently and is unclear.
- “The error! Reference source not found” maybe appearing due the Ms Word version mode used to save the manuscript. We have changed into compatibility mode to avoid this from happening.
- Lines 239-240 – In “Overall, 52 minutes was spent on a typical day on recreation-related activity. Male participants spend more time (58.7 minutes) compared to females (42.5 minutes) per day”, authors should provide the recommended recreation-related activity levels for reference.
- Changes have been made to provide more clarity. We are describing the average amount of time spent on recreation-related activity. For adults aged 18-64 years, the WHO recommendation uses a combination of all physical activity to determine levels of physical activity other than a single physical activity. We have included the overall level of physical as introductory sentence when reporting the level of physical activity among the participants.
Reviewer 4:
- The acronym ‘CD’ (chronic diseases of lifestyle) should be spelled out the first time it is introduced.
- We have made changes to improve consistency. I have used the phrases and abbreviations when I first introduced CDL, WHR, BMI, WHO among other abbreviations.
- Line 28-32 – Specify if this information on the epidemiologic transition applies to youth in Kenya or youth worldwide, since the supporting references focus on Kenya.
- Citations from the 2 MoH reports and 1 from the WHO highlighting the issue in Kenya.
- Lines 35-39 – This is insightful information. It would be interesting to know what contributes to this dietary shift in Kenya.
- Suggestion has been implemented.
- Regarding the study population, how were participants recruited? Were incentives provided to participate in the study?
- We have made changes. We have highlighted the steps in recruitment in the sampling section. Incentives were also provided and has been incorporated in the manuscript in the data collection section.
- There is insightful discussion of the sampling strategy.
- Line 104 – In “Specifically, those who met the inclusion criteria”, what was the inclusion criteria? Were there exclusion criteria?
- We have included the inclusion and exclusion criteria under the study population.
- Lines 119-120 – Check wording and punctuation (consider comma after ‘needed’) in “Where needed the researcher and assistants was available to assist the participants”.
- We have made changes. We have inserted a comma after the word “needed”.
- Line 136 – The acronym ‘WHR’ should be written in full the first time it is introduced.
- We have made changes. The full name has been inserted where the acronym is first introduced throughout the manuscript.
- Lines 165-166 – This sentence is unclear – “The socio-demographic characteristics of the study participants are presented in Error! Reference source not found..”. It is not clear what this statement is referring to.
- “The error! Reference source not found” maybe appearing due the Ms Word version mode used to save the manuscript. We have changed into compatibility mode to avoid this from happening. The statement was referring to table 1.
- Lines 180-181 – Again this sentence is unclear – “The overall, mean total body fat percentage for males was 14.1% and 30.4% for females Error! Reference source not found..”.
- “The error! Reference source not found” maybe appearing due the Ms Word version mode used to save the manuscript. We have changed into compatibility mode to avoid this from happening.
- Lines 196 and 198; 210-211; 258-269; 261-262, 294 – The statement “Error! Reference source not found” is presented frequently and is unclear.
- “The error! Reference source not found” maybe appearing due the Ms Word version mode used to save the manuscript. We have changed into compatibility mode to avoid this from happening.
- Lines 239-240 – In “Overall, 52 minutes was spent on a typical day on recreation-related activity. Male participants spend more time (58.7 minutes) compared to females (42.5 minutes) per day”, authors should provide the recommended recreation-related activity levels for reference.
- Changes have been made to provide more clarity. We are describing the average amount of time spent on recreation-related activity. For adults aged 18-64 years, the WHO recommendation uses a combination of all physical activity to determine levels of physical activity other than a single physical activity. We have included the overall level of physical as introductory sentence when reporting the level of physical activity among the participants.
Round 2
Reviewer 3 Report
Thank you for addressing my comments and questions.
Author Response
Thank you.
Reviewer 4 Report
The paper is improved from the first version and the authors should be commended for their work. Below are some remaining items to review.
- Line 28- Review the positioning of the graphical abstract at the beginning of the paper. Also, the font size in the figure is not uniform. The word ‘chronic’ in graphical abstract (first text box) is falling out of view.
- Line 112 – Check wording in “being a registered students”.
- Lines 132-133 – There is a thoughtful gesture and good mention that “Each
- participant was gifted with a bag of fruit after completing data collection as a token of appreciation for their participation”.
- Line 281 – Check spacing in “( 3000 METs)”.
- Lines 336-339 – In “However, participants who consume fruit daily or who consume vegetables daily and met the WHO recommendation of consuming at least five servings of fruit and/or vegetables per day were 30.0%, 48.0%, and 36.0% more likely to be overweight, respectively”, this is a curious finding. It seems to state that meeting WHO recommendation of consuming at least five servings of fruit and/or vegetables per day is associated with increased likelihood to be overweight. Would this be due to increased caloric intake overall? Authors should explain.
- Line 339 – Check mention of “(Error! Reference source not found.)”. Also check for this in lines 211, 226, 241, 243, 245, 303, 306.
- Good discussion of policy implications.
Overall, this paper is clearer, more detailed, and much improved. Authors should attend to some remaining clarifying questions as stated above. Authors should also careful check for the instances of “(Error! Reference source not found.)” found throughout the paper. Once these
Author Response
Dear Editor,
RE: RESPONSES TO REVIEWERS COMMENTS/SUGGESTIONS
Please find enclosed our revised manuscript. We thank the reviewers for their constructive comments on our manuscript. We have fully addressed comments in the revised the manuscript and we believe these revisions have resulted in a significantly improved manuscript.
Below we outline how outline how we have handled each of the reviewers’ comments. We reiterate each suggestion in italics.
- Line 28- Review the positioning of the graphical abstract at the beginning of the paper. Also, the font size in the figure is not uniform. The word ‘chronic’ in graphical abstract (first text box) is falling out of view.
- We have made changes to the graphical abstract.
- Line 112 – Check wording in “being a registered students”.
- We have made changes in the sampling section.
- Line 281 – Check spacing in “( 3000 METs)”.
- We have made changes under the behaviourial risk factors.
- Lines 336-339 – In “However, participants who consume fruit daily or who consume vegetables daily and met the WHO recommendation of consuming at least five servings of fruit and/or vegetables per day were 30.0%, 48.0%, and 36.0% more likely to be overweight, respectively”, this is a curious finding. It seems to state that meeting WHO recommendation of consuming at least five servings of fruit and/or vegetables per day is associated with increased likelihood to be overweight. Would this be due to increased caloric intake overall? Authors should explain.
- We have made changes. Included further details that could attribute to the finding.
- Line 339 – Check mention of “(Error! Reference source not found.)”. Also check for this in lines 211, 226, 241, 243, 245, 303, 306.
- We have made changes and resolved the error throughout the results section.
Many thanks.
Kind Regards,
